# Bioengineering Liver Organoids for Diseases Modelling and Transplantation

**DOI:** 10.3390/bioengineering9120796

**Published:** 2022-12-13

**Authors:** Junzhi Li, Jing Chu, Vincent Chi Hang Lui, Shangsi Chen, Yan Chen, Paul Kwong Hang Tam

**Affiliations:** 1Department of Surgery, School of Clinical Medicine, The University of Hong Kong, Hong Kong SAR, China; 2Li Dak-Sum Research Centre, The University of Hong Kong, Pokfulam Fu Lam, Hong Kong SAR, China; 3Department of Mechanical Engineering, The University of Hong Kong, Pokfulam Road, Hong Kong SAR, China; 4Faculty of Medicine, Macau University of Science and Technology, Macau SAR, China; 5Division of Paediatric Surgery, Department of Surgery, Queen Mary Hospital, Li Ka Shing Faculty of Medicine, The University of Hong Kong, Hong Kong SAR, China

**Keywords:** bioengineering, liver organoids, liver disease models, transplantation

## Abstract

Organoids as three-dimension (3D) cellular organizations partially mimic the physiological functions and micro-architecture of native tissues and organs, holding great potential for clinical applications. Advances in the identification of essential factors including physical cues and biochemical signals for controlling organoid development have contributed to the success of growing liver organoids from liver tissue and stem/progenitor cells. However, to recapitulate the physiological properties and the architecture of a native liver, one has to generate liver organoids that contain all the major liver cell types in correct proportions and relative 3D locations as found in a native liver. Recent advances in stem-cell-, biomaterial- and engineering-based approaches have been incorporated into conventional organoid culture methods to facilitate the development of a more sophisticated liver organoid culture resembling a near to native mini-liver in a dish. However, a comprehensive review on the recent advancement in the bioengineering liver organoid is still lacking. Here, we review the current liver organoid systems, focusing on the construction of the liver organoid system with various cell sources, the roles of growth factors for engineering liver organoids, as well as the recent advances in the bioengineering liver organoid disease models and their biomedical applications.

## 1. Introduction

An organoid is an in vitro three-dimensional (3D) miniaturized and simplified version of an organ that shows realistic micro-anatomy to partially mimic the structure and function of a native organ. Organoids hold great promise for disease modelling, drug screening, cancer mechanism investigation, and regenerative medicine [1,2,3,4,5,6,7]. Organoids have been employed as a novel tool to study the development of various organs and tissues including brain [5], lung [8], liver [9], stomach [10], intestine [11], pancreas, and kidney [12]. The organoid culturing system contains different components including essential nutrients for supporting cell growth/proliferation, growth factors for inducing specific cell differentiation, and biomaterials for providing mechanical support and biochemical environment [13,14]. Versatile organoid culturing techniques have been developed to generate organoids with destined phenotype and controlled morphology to uncover the development of human-specific organs [15,16].

Liver organoids have emerged as versatile and helpful tools to elucidate the normal and abnormal development of liver in healthy and disease conditions. Research groups have successfully generated liver organoids in biomatrix gel from human fetal liver progenitor cells [17], induced pluripotent stem cells (hiPSC) [18] and pluripotent stem cells (PSC) [19], endothelial and mesenchymal cells [20]. Liver organoids derived from normal livers [21] and diseased livers [22] have also been reported. In addition to biological cues such as growth factors [23], secreted proteins [24], and molecular signals [10,25] that govern the self-organizing and differentiating processes of cells for organoid formation, biochemical and biophysical properties of the biomatrix gel are also crucial for regulating cell proliferation/differentiation and their ultimate morphology [26]. Furthermore, burgeoning fabrication techniques (involving 3D printing, bioprinting, microfluidic chip, light patterning, etc.) have provided organoids with more sophisticated shapes and functions to meet different challenges in biomedical fields. In this review, we will discuss the evolution of liver organoid bioengineering and its applications in biomedicine and liver transplantation. 

## 2. Bioengineering Liver Organoids

The significant events of bioengineering liver organoids are illustrated in the timeline of Figure 1. The 3D spheroidal aggregate derived from rat liver cells could be the beginning of the journey of liver organoid development [27]. Long-term cultures of rat hepatocytes were achieved by using liver biomatrix [28] and a coculture system consisting of fibroblasts and hepatic parenchymal cells could be applied in forming 3D liver tissue [29]. The 1980s witnessed the start of culturing 3D liver structures and laying the groundwork for the emergence of liver organoids in the 1990s. Biodegradable scaffolds [30], hydrogels [31], and polymeric microcarriers [32,33,34] had been employed to engineer liver organoids, which could be the starting of bioengineering liver organoids, meanwhile, coculture systems were developed to reconstruct liver organoids in the period of 1990s [31,35]. In the 2000s, more and more techniques such as bioreactor [36], light patterning [37,38], and bioprinting [39] were induced for engineering a liver organoid. Extracellular matrix combined with polymeric scaffolds were used to improve the survival and morphological integrity of hepatocytes [40]. When it came to the 2010s, advanced liver organoids with a vascular network [41] and gene-modified liver organoids [42] were presented in the laboratories. The long-term culture and expansion of liver organoids play a key role in constructing reliable and replicable liver development and disease models, which was achieved by Hans Clevers’ group [23,24]. Liver cancer organoid models that can be used to understand liver cancer biology and develop therapeutics have also been established in the 2010s [4]. Currently, in the 2020s liver organoids were extended to biliary atresia [43] and virus-infected models [44]. The functional hydrogels [45] and the advanced printing technique [46] have attracted more and more attention to their application in engineering liver organoids.

What are the compositions of a liver organoid and how do we engineer a liver organoid? A bioengineering liver organoid platform is a complex system involving various basic biological research and bioengineering disciplines (Figure 1, bottom). This part reviews the cell sources, biological agents, biomaterials, and fabrication techniques that have been employed in the generation of liver organoids.

### 2.1. Liver Organoids Derived from Different Cell Sources

Liver primary cells which express a surface receptor Leucine-rich repeat-containing G protein-coupled receptor 5 (LGR5) were identified to be self-renewable and a single LGR5-expressing cell was able to produce liver organoids of functional cholangiocytes and hepatocytes [47]. Transplantable liver organoids have been generated from long-term clonally expanded LGR5+ cells using an R-Spondin1 (Rspo1)-based culture medium [48]. Large-scale expansion of LGR5+ human liver cells was achievable using spinner flasks to improve oxygenation of the culture resulting in a 40-fold cell expansion after 2 weeks of culture [47]. Long-term culture of feline bile duct tissue-derived liver organoids retained the characteristics of adult liver stem cells [49]. Primary bile duct cells have been expanded in vitro to form 3D liver organoids and these cells also can differentiate into functional hepatocyte cells [23]. All these findings indicated that the liver LGR5+ primary cells can maintain their “stemless” properties after long-term culture and expansion and can be induced to form liver organoids. 

Induced pluripotent stem cells (iPSC) are capable of differentiating into many different cell types, and iPSC-derived liver organoids can well replace the conventional liver tissue-derived organoid systems [50]. Protocols have been established to induce iPSC to differentiate into liver organoids of cholangiocytes and hepatocytes [51,52,53,54,55]. Typically, iPSCs were cultured in a medium containing bone morphogenetic protein 4 (BMP4), fibroblast growth factor-4 (FGF4), and B-27 supplements to induce differentiation of iPSC into hepatoblast-like cells, and subsequently cultured with epidermal growth factor (EGF) in matrigel to encourage cholangiocyte differentiation [56]. Careful selection of extracellular matrix (ECM) is a key step toward cholangiocyte differentiation from human iPSC: laminin 411 and laminin 511 have been shown to promote cholangiocyte differentiation [56]. The construction of iPSC-derived cholangiocytes for disease modelling (Alagille syndrome) and drug validation (cystic fibrosis drug VX809) have been reported [57]. There are two steps to generate a liver organoid from iPSCs. Initially, iPSCs are differentiated into hepatic endoderm and progenitor cells using specific protocols under a 2-dimension (2D) culture environment. Thereafter, the cells will be seeded in a 3D culture to construct a complex 3D organoid. However, to engineer a transplantable and functional in vitro organoid, the vasculature plays a significant role in producing suitable engraftment with biological functions. Takebe et al. reported a protocol for generating liver buds from human iPSCs and investigated the vascularization and maturation of implanted liver buds in immuno-deficient mice [58].

Multiple cell-type liver organoid systems allow more sophisticated structures to be engineered with the vasculature and biliary system to mimic a native liver. Vascularization is a crucial step toward organogenesis, which is essential for oxygen and nutrition transportation and distribution. A liver organoid platform involving iPSCs, mesenchymal stem cells (MSCs), and human umbilical vein endothelial cells (HUVECs) was employed to investigate the paracrine effects of stem cell therapies for regenerative medicine [25,59]. Liver organoid models containing hepatocytes and cholangiocytes derived from iPSCs were employed to study human genetic disorders [60]. 

### 2.2. Biological Factors for Directing Liver and Biliary Organoid Formation

Biological factors play a pivotal role in the regulation of cell expansion, differentiation and self-organization in in vitro organoid cultures. As liver is developed from embryonic endoderm, which also give rise to the endoderm tissues in pancreas and gut tube [61]. The biological factors implicated in some of the important molecular signaling pathways underlying the proliferation/migration/differentiation of endoderm such as Noggin, epidermal growth factor (EGF), transforming growth factor beta (TGF-β), hepatocyte growth factor (HGF), fibroblast growth factor (FGF), R-spondin 1 (a Wnt pathway potentiator), TGF-β inhibitor (A83-01), a cAMP pathway agonist forskolin (FSK), and retinoic acid (RA) have been used in the generation of organoids of endoderm origin including liver organoid [62]. A medium containing these factors supported the long-term culturing of liver organoids for genetic manipulation [42].

### 2.3. Biomaterials for Engineering Liver Organoids

The ultimate goal of liver tissue engineering is to create a transplantable artificial liver for transplantation to patients with end-stage liver failure. The liver ECM can act as a scaffold supporting cell survival, proliferation, and differentiation [63]. A 3D environment influences cellular behaviors via the interactions between cells and the ECM, and an absence of cell-ECM interaction could lead to a specific type of cell apoptosis called anoikis [64]. Cell migration and differentiation are also influenced by the physical properties such as the rigidity, adhesion, confinement, topology, and biochemical signals of the ECM [65,66]. A variety of biomaterials have been investigated and employed to reconstruct an appropriate microarchitecture with suitable stiffness and liver-specific ECM protein signals, for supporting liver organoid in vitro expansion. 

Traditional polymers such as polyhydroxyalkanoate and poly(lactic-co-glycolic acid) (PLGA) polylactic acid (PLA), polycaprolactone (PCL), and poly(ethylene glycol) diacrylate (PEGDA) have been employed for liver tissue scaffold fabrication [67,68]. These polymeric materials have good long-term biocompatibility, quality mechanical properties, and biodegradable feature for liver organoid expansion. However, most of these polymers have a high mechanical stiffness that is not applicable for 3D liver organoid culture. In contrast, hydrogels, with their soft mechanical properties, become the best candidate materials to mimic the mechanical environment of soft tissue like liver [69]. 

Alginate is a natural polysaccharide, which is widely used as biomedical hydrogel scaffolds due to its good biocompatibility and affordable price [70]. It can be crosslinked by divalent cations to provide a stable architecture supporting liver organoid growth and proliferation [71]. Alginate is a flexible material for a number of fabrication and modification techniques, which enables organoid encapsulation for biomedical applications. For example, microporous alginate scaffolds were fabricated to promote the development of infant hepatic cells into hepatic tissue for drug screening [72]. Gelatin methacryloyl (GelMA), a hydrogel that can be photo crosslinked, is typically prepared by a direct reaction of gelatin and methacrylic anhydride (MA). Many preparation methods were developed for its widespread applications in the biomedicine [73]. Organoid-laden complex scaffolds which are composed of GelMA, polyisocyanopeptides (PIC), and laminin-111 enable long-term expansion of human liver organoids up to 14 passages with a proliferation rate comparable to commercial Matrigels [46]. Hyaluronic acid (HA) and collagen are the main components of native organ ECM, which have been widely used in regenerative medicine and tissue engineering scaffolds. A host-liver colorectal-tumor organoid model was established using HA-based microcarriers loaded with ECM elements and liver-specific growth factors, which demonstrated a possible application of the HA-based hydrogel for liver tumor model [74,75]. Condensed collagen fibril scaffolds have been employed to encapsulate liver organoids into a transplantable hepatic tissue, which provides a microenvironment enhancing the cell-cell and cell-ECM interactions, and maintaining liver-specific functions of the organoids [76]. The native organ decellularization matrix could be an implantable scaffold for transplantable organoid tissue in vitro reconstruction. The whole organ decellularization matrix was used to produce vascularized liver organoids [41], and various decellularized ECM (dECM) liver organoid systems were reported [77,78,79].

Compared with the above-mentioned hydrogels, hybrid hydrogels can address some of the limitations of using a single component matrix for liver organoid culture. To date, the most prevalent commercial matrix is Matrigel, a basement membrane extracellular matrix, that mainly consists of collagen IV, nidogen, perlecan, and a high content of glycosylated molecules [80]. Matrigel has been widely used for liver organoid expansion (for review see [50,81]). The reasons for the widespread use of commercial Matrigel for liver organoid generation are due to its (i) good biocompatible properties; (ii) stable thermal-sensitive gelation properties; and (iii) optically transparent properties for real-time monitoring of organoid growth. The clinical applicability of Matrigel is, however, severely limited by the variability in its composition and the presence of xenogenic contaminants. Other hybrid-hydrogel scaffolds have also been used to fabricate organoid-laden scaffolds, which diversify the materials for the organoid generation [45,82].

### 2.4. Current Fabrication Techniques for Engineering Liver Organoid

With the development of material processing technology, large-scale fabrication of liver organoids for transplantation is not unachievable. An engineering technique for building implantable liver organoids should fulfill the criteria not only in the material processing aspect but also in the biological aspect. Three-D printing technology allows the creation of physical objects from a geometrical representation by successive addition of materials. Meanwhile, 3D bioprinting, a combination of 3D printing with biological content, has emerged as a powerful tool for the fabrication of biomimetic and implantable scaffolds [83,84], which is also used for the production of organoid-laden structures [85]. Manon et al. reported an extrusion-based bioprinting approach using GelMA hydrogel as a printing ink to construct liver tissue-derived epithelial organoids [86]. The viability of printed organoids in GelMA hydrogel could be maintained at not less than 88% up to 10 days, which enabled the hepatotoxin exposure experiment to be conducted. A nozzle-free volumetric bioprinting technique has been developed to fabricate gelatin-based scaffolds encapsulating hepatic organoids [46]. This report illustrated that ammonia detoxification was modulated in the printed liver-specific organoid using a perfusion system feeding with NH_4_Cl. 

Suspension culture is a well-developed technique for 3D organoid expansion for the scalable production of organoids. Researchers have developed a two-step protocol for the large-scale production of liver organoids from human iPSCs, in that homogeneous and uniform-sized human embryoid bodies (hEBs) were first generated and the hEBs were then cultured in a suspension culture platform to form liver organoids [87]. Saskia and his colleagues have developed a protocol for the scalable production of hepatocytes and hepatic organoids; furthermore, their protocol also included a cryopreservation method to store intermediate hepatic organoids for the efficient production of a large number of organoids [88]. 

Organ on a chip is an effective in vitro platform to reflect human liver-specific functions, as well as the complex process of detoxication and bile production. Organ on a chip platform can provide a dynamic microenvironment by manipulating the compositions of the perfusion fluids. To mimic liver functions and the native liver microenvironment, a chip with a specific design can be fabricated for a variety of in vitro studies. In a microfluidic 3D human liver sinusoid, faster albumin and urea responses were observed under continuous perfusion, allowing drug screening on a liver-on-a-chip platform [89]. Microfluidic organ-on-a-chip devices were designed and produced to study the in vitro liver metabolism and assess cardiac safety of drugs (e.g., clomipramine) through a co-culture model of liver and cardiac organoids on the chip [90]. Using this multi-organoid platform, the urea synthesis was measured for evaluating the liver-specific function of liver organoids and the expression of liver-specific CYP450 enzyme genes. 

## 3. Biomedical Applications of Bioengineering Liver Organoids

Liver disease leading to liver failure is a worldwide issue. The ultimate goal of liver bioengineering is (i) to develop appropriate hepatic models to reveal the pathological mechanism of liver disorders, and (ii) to create transplantable liver tissue to replace the damaged one. In this section, bioengineering liver organoid disease models, cancer models, and virus-infected models, as well as transplantable liver tissue derived from organoids will be reviewed (Figure 2). 

### 3.1. Liver Organoid Disease Models

Neonatal cholestatic disorders like biliary atresia (BA), a leading cause of paediatric liver failure, impact the cholangiocytes of the intrahepatic and extrahepatic biliary tree [91,92]. Bile flow is obstructed leading to severe liver fibrosis and impaired bile duct system. BA has caused suffering in thousands of families with disease incidences of 0.05 to 0.14% in live births. Although Kasai portoenterostomy can effectively improve early clinical outcomes, over 70% of BA patients have to ultimately undergo liver transplantation to survive [93]. Its etiology remains to be fully revealed, but it is known that genetic mutations, exposure to hepatotoxin, and infection by viruses could contribute to the cause of BA [92]. Our group generated liver organoids from the liver of BA patients and BA mice (Rhesus rotavirus A-infected mice) and showed that BA patient- and BA mouse- liver organoids displayed BA-specific aberrant morphology, indicating that human liver organoids can be a good human proxy for patho-mechanistic study for BA [43]. Specifically, liver organoids from BA patients and BA mice exhibited aberrant morphology and disturbed apical-basal organization; a defective cholangiocyte development and altered beta-amyloid-related gene expression in BA organoids; β-amyloid peptide (Aβ) accumulation in BA livers and exposure to Aβ induced the BA-specific aberrant morphology in control organoids. The delayed epithelial development and barrier function in BA organoid model was revealed by Bezerra’s research group [94]. 

### 3.2. Liver Organoid Cancer Models

The incidence of human primary liver cancer (PLC), which includes hepatocellular carcinoma (HCC), cholangiocarcinoma (CC), and combined HCC/CC (CHC) tumors, is rising likely due to increased/high incidences of hepatitis, diabetes, obesity, etc., and PLC has become one of the most common lethal malignancies worldwide [95,96]. The organoid model derived from PLC liver is a novel platform for liver cancer biology study, drug screening and personalized medicine [4]. PLC liver-derived organoids were expanded using human PLC tissue according to an optimized protocol which was developed by Laura and coworkers [23,42]. PLC liver-derived organoids recapitulate the expression profile and the histological architecture of human PLC tissues, and are now termed PLC-specific tumoroids [4]. The patient-specific sensitivity of anticancer compounds was evaluated using this PLC-derived organoid system. PLC-derived organoids with the HCC feature demonstrated a high sensitivity to anticancer drugs like Gemcitabine, AZD8931, SCH772984, Taselisib, and Dasatanib. CC and HCC-derived organoids were transplanted into NSG mice (NOD SCID gamma mouse) to establish an in vivo model for investigating the effect of ERK inhibitors such as SCH772984 on PLC. Modeling cancer initiation can provide an effective tool to monitor the histological changes during the early stage of cancer development. It is known that the overexpression of c-Myc may cause HCC [97,98,99]. Lulu and coworkers have reported an approach to mimic liver cancer initiation by overexpressing c-Myc in genetically reprogrammed human hepatocyte (hiHeps) organoids [100]. The whole-transcriptome analysis confirmed that activation of the c-Myc pathway in the reprogrammed hiHeps organoids. Furthermore, it was found that mitochondrion-associated endoplasmic reticulum (ER) membranes (MAMs) presented a disorganized morphology and a high level of ER apposition to mitochondria, and a higher degree of mitochondrial Ca^2+^ uptake in the c-Myc organoids. RAS^G12V^-transformed hiHep organoids (RAS organoids), which faithfully recapitulated the characterizations of RAS-mutated liver cancer, were employed to study the driving effect of RAS on human intrahepatic cholangiocarcinoma (ICC) formation from human hepatocytes. Organoids derived from PLC tissues and genetically engineered liver organoids derived from reprogrammed human hepatocytes represent two available strategies for modelling human liver cancer initiation and development, which will yield insights in cancer prevention and treatment targets.

### 3.3. Virus-Infected Liver Organoid Models

The current pandemic of coronavirus disease 2019 (COVID-19), caused by the severe acute respiratory syndrome coronavirus 2 (SARS-CoV-2), a single-strand positive-sense RNA virus, has become a worldwide threat that leads to over 600 million persons infected and at least 6 million deaths according to the World Health Organization (WHO) until now [101]. Meanwhile, increasing COVID-19-related liver dysfunction cases have been reported [102]. Organoids as promising in vitro models offer an affordable 3D self-organized cellular platform to researchers for the investigation of human infectious diseases and host-microbe interactions [103]. 

Lui and coworkers have reported a human liver organoid-derived intrahepatic bile duct cell platform for conducting the comparative study of SARS-CoV-2 and SAR-CoV infectivity and replication rate [104]. It was shown that cholangiocytes were susceptible to SARS-CoV-2 infection, and also supported efficient viral replication. Furthermore, SARS-CoV-2 replication was shown to be much higher than SARS-CoV. Taken together, these suggested direct cytopathic viral damage to be a mechanism for SARS-CoV-2 liver injury. 

Virus-infected organoids emerged as a platform for testing antiviral drugs. Hepatitis E virus (HEV), a positive-stranded RNA virus, often causes the epidemic of acute hepatitis (hepatitis E) in subtropical countries [105]. As hepatitis viruses usually infect a narrow range of hosts, human liver-derived organoids have been employed to study the virus-host interactions between the HEV and the liver cells [81]. This report showed that human liver tissue-derived organoids supported HEV infection, offering a novel tool for conducting a genome-wide transcriptomic analysis of HEV infection and antiviral drug screening in the HEV-infected organoid system. It was found that the organoids have a vigorous host response to HEV along with an upregulation of the ISGs gene. Nie et al. have reported human iPSCs-derived functional liver organoids recapitulated the hepatitis B virus (HBV)-host interactions in vitro [106]. Hepatic dysfunction has been observed in HBV-infected organoids along with down-regulation of hepatic genes including *ALB*, *HNF1A*, *RBP4*, *G6PC*, *CYP3A4*, *CYP3A7,* and *CYP2C9*. Recently, scientists have reported a coculture system composed of the liver organoid and CD8+ T cells using a microfluidic technique to model T-cell immunity against the hepatitis C virus [107].

### 3.4. Limitations of the Liver Organoid Models

There are limitations to using liver organoids as disease and cancer models. Compared to traditional 2D culture systems, organoid culturing required costed biological agents and a more complicated environment. The liver organoid model is still in the infant state. Only limited functionality and characteristics can be replicated in the models, even though some culture coculture liver organoid platforms exist. The architecture of the bioengineered liver organoids highly relies on the stiffness of supporting biomaterials. However, there are a very limited number of commercial products that can be used for liver organoid culture and manipulating the morphology of liver organoids, making liver organoids a single spherical shape. From the point of view of cell mechanics, the single morphology of liver organoids can hardly mimic the cell-cell mechanical interactions. 

### 3.5. Transplantable Liver Organoid Tissue

The ultimate goal of liver tissue engineering is to create transplantable liver tissue to replace the damaged part and restore hepatic functions. Vascularization is a critical step toward a transplantable liver organoid-derived tissue (for a review see [108]). A transplantable liver organoid tissue with a sandwich structure composed of two layers of collagen fibrils and inner clusters of hepatocytes was designed and fabricated [76]. In the fabricated organoid-laden scaffolds, microvascular networks were formed after transplantation, which suggested that the interaction between the host and graft may facilitate blood vessel formation. Limitations of using organoid-derived tissues or cells in clinical applications include poor control of morphology and composition and insufficient amounts of tissues/cells. To address these problems, a high-throughput technique was developed to generate transplantable liver organoids in a scalable and reproducible way [109]. The bile ducts may be removed during an operation for a tumor, post-traumatic/inflammatory, or congenital problem such as biliary or choledochal cyst biliary reconstruction to repair the damaged or missing biliary tissue may be achieved with intestinal or preferably biliary conduits. However, the shortage of healthy donors poses an obstacle to like-for-like biliary reconstruction. Recently, scientists have advanced the field by successfully reconstructing the gallbladder wall and bile duct using extrahepatic cholangiocyte organoids in mice [21]. 

Transplantation of liver organoids into a liver can be regarded as an organoid delivery process. Apart from direct transplantation of liver organoids to the recipient’s liver, other routes of liver organoid delivery to liver have also been tested. Tomonori and coworkers have conducted an animal study in transplantation of porcine fetal liver-derived allogeneic organoids to the 10-day-old piglet liver via the portal vein [110]. Their results indicated that it is a safe procedure to deliver iPSC-derived and liver fetal tissue-derived liver organoids through the portal vein with a ligation of the venous catheter, which can minimize the risk of extrahepatic translocation of the liver organoids. 

## 4. Prospect and Summary

Bioengineering liver organoid technology provides a useful platform for studying liver development, virus infection of liver, liver disease and cancer, and for producing transplantable liver tissues. The liver serves as a chemical factory of the human body. However, the liver organoid system can only recapitulate part of physiological functions and structure of a liver. A more sophisticated liver organoid system containing multiple liver cell types that could mimic the full physiological functions of a liver is an ultimate goal of tissue bioengineering. To generate and maintain such a complex multiple cell types organoid system, a deeper understanding of the compositions of the biomolecules in the liver ECM that provides the biological cues and optimal mechanical/physical properties is essential for the future development of biomaterials for the generation of the next generation complex organoid system. Therefore, the synthesis/discovery of advanced biomaterials that could provide optimal mechanical and physical support and biological cues for regulating the 3D liver organoid growth and expansion will be critical for the future development of liver organoids. 

This review summarizes the advantages and obstacles of bioengineering liver organoid techniques and their applications in biomedical and clinical research. The field has witnessed a rapid growth of interest in engineering liver organoids with specific features and functions for various applications. Recent advancements in organoid culturing and fabrication technologies have promoted the complexity of the liver organoid systems for applications in liver disease patho-mechanistic study, liver cancer modelling, virus-infection modelling, and the production of transplantable liver tissues. 

## Figures and Tables

**Figure 1 bioengineering-09-00796-f001:**
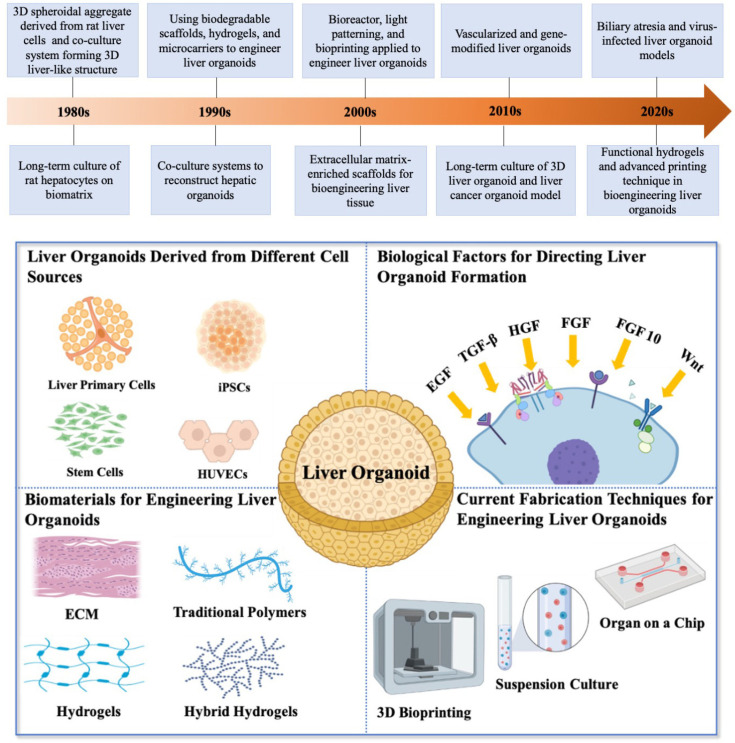
Timeline of the important events of bioengineering liver organoids (top): the 1980s witnessed the evolution of the 3D liver-like spheroidal aggregate [27,28,29]; the 1990s witnessed the development of liver organoids [30,31,32,33,34,35]; various techniques and biomaterials had been employed in bioengineering liver organoids in the 2000s [36,37,38,39,40]; the 2010s witnessed the development of functional liver organoids and their long-time culture and cancer models [23,24,41,42]; functional hydrogels and advanced printing technique have emerged in bioengineering liver organoids [45,46] and the liver organoids were exposed to biliary atresia and virus-infected models [43,44]; and the diagrammatic sketch of bioengineering liver organoids with different cell sources, biological factors, biomaterials, and fabrication techniques (bottom). iPSCs: Induced pluripotent stem cells; HUVECs: human umbilical vein endothelial cells; EGF: epidermal growth factor; TGF-b: transforming growth factor beta; HGF: hepatocyte growth factor; HGF: fibroblast growth factor.

**Figure 2 bioengineering-09-00796-f002:**
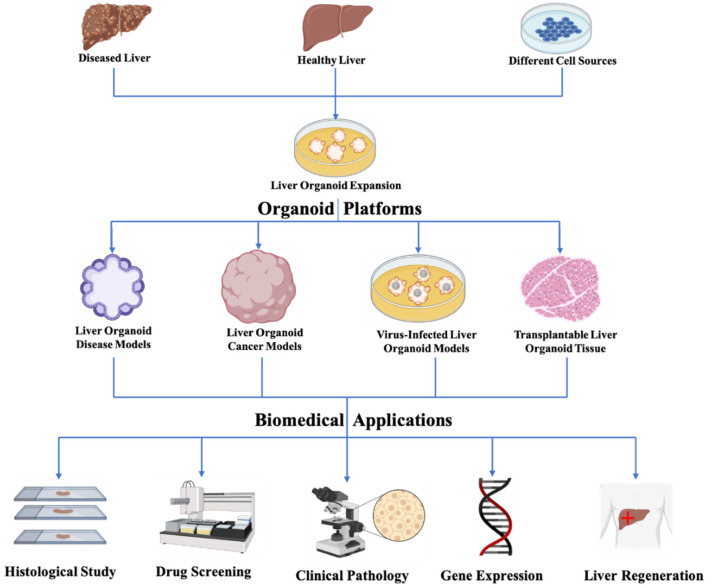
Bioengineering liver organoids derived from various sources to construct disease models, cancer models, virus-infected models, and transplantable organoid tissue for their application in histological study, drug screening, clinical pathology, gene expression analysis, and regeneration.

## Data Availability

Not applicable.

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
