# Peer review of "Bioengineering Liver Organoids for Diseases Modelling and Transplantation"

_bioengineering, 2022, doi:10.3390/bioengineering9120796_

Round 1

Reviewer 1 Report

This manuscript, by Junzhi Li and coauthors, is a a comprehensive review article on the recent advancement in the bioengineering of liver organoids. The authors review the state of the art in the field and discuss biotechnological aspects (i.e., the "construction" of liver organoid systems) as well as implications of liver organoid research for biomedical applications. The manuscript is interesting and timely. I do recommend its publication in Bioengineering.

Author Response

We thank you for your consideration and recommendation. We are so glad to hear the positive comments on our review “Bioengineering Liver Organoids for Diseases Modelling and Transplantation”. We hope we can do better in publishing this review paper on bioengineering.

Reviewer 2 Report

This review type manuscript unfortunately does not contain much in the way of new information on this topic, over and above some excellent review articles that have already been published in recent years

Plus, the speculation in the manuscript that large volumes of liver tissue can be bioengineered via an organoid type approach seems misplaced as there remains some well known limitations and unmet challenges with respect to bioengineering organoids 

Please see my comments to the authors

There are some limitations with the manuscript in it's current form for which I will make point by point comments

1) First, there have been a number of review articles published on liver organoids in recent years, particularly as to how they are so called bioengineered (along with the current research and potential clinical applications). This is one of the recent better reviews Liver Organoids: Recent Developments, Limitations and Potential - PubMed (nih.gov) Plus reference number 21 in the Reference list for your manuscript (Nuciforo et al)- provides a good summary of the potential clinical applications of liver organoids for some of the types of liver disease (along with the limitations and challenges). Hence there is not much in the way of new information in your manuscript

2) The potential to transplant large volumes of liver organoid tissue remains very much a long range goal, with instead the potential to transplant smaller volumes of liver organoid tissue being a far more realistic possibility. You need to make this distinction in the manuscript and instead focus on where the research may be heading for the use of smaller amounts of organoids as a therapeutic modality in the treatment of certain types of liver diseases.

3) Based on the above comments the manuscript is best rewritten

Author Response

Thanks for reading our manuscript and providing comments. In this review, we have included the current liver organoid systems, focusing on the construction of the liver organoid system with various cell sources, the roles of growth factors for engineering liver organoids, as well as the recent advances in the bioengineering liver organoid disease models, and their biomedical applications. It is undeniable that many excellent review articles have reviewed the development of different organoid systems likely due to the study of organoids becoming an increasingly hot issue. However, few review articles are providing an exhaustive collection of published liver organoid papers from a point of biomedical engineering view. Bioengineering liver organoids are facing many limitations and challenges, which may pose a barrier to its progress, but this decade also witnesses the development of various organoid platforms for understanding organ development. Therefore, we hope our review can make a contribution to researchers in some way by providing a comprehensive review of the current progress of bioengineering organoids.

Response to comment 1:

We have gone through the review “Liver Organoids: Recent Developments, Limitations and Potential” which has comprehensively reviewed the development, limitation, and application of liver organoids in the aspect of biological medicine. However, our review is focusing on how to engineer a liver organoid with various biological components and advanced fabricating techniques. We can provide the audience with more information on bioengineering liver tissues and organoids.

Response to comment 2:

Any research should have long-term gold and short-term gold. As The ultimate goal of liver tissue engineering is to create transplantable liver tissue to replace the damaged part and restore hepatic functions. Therefore, reviewing the application of liver organoids in transplantation is necessary

Response to comment 3

Thanks for your comments and advice. The revised manuscript was submitted to bioengineering.

Reviewer 3 Report

The authors have made an interesting attempt at “Bioengineering Liver Organoids for Diseases Modelling and Transplantation” The manuscript is interesting; however, the authors need to justify the scientific writing manuscript. Some of the general comments are provided below:

1.     The article is very dense and there is almost too much detail. Perhaps adding a few research questions early on would help.

2.     A table showing the periodic development of the Liver organoids would be easy to follow.  

3.     Is there any study related to Hepatitis C for using liver organoids as a disease model or transplantable?

4.     The authors should include a separate part at the end as “limitations of the model”. 

Author Response

Response to comment 1:

 The dense structure of the article may provide key knowledge and an overall look at bioengineering liver organoids to audiences within a relatively short reading time. A few questions were added at the beginning of section 2.0 to give a preview of this section (see section 2. , the first sentence page 4).

Response to comment 2:

A time bar of important events in the development of liver organoids from a bioengineering point of view was added in the top revised Figure 1.

Response to comment 3:

A paper named “Modelling T-cell immunity against hepatitis C virus with liver organoids in a microfluidic coculture system” was just published online. We have included this Hepatitis C-infected liver organoid model in Section 3.3 Virus-Infected Liver Organoid Models (see page 16 in the first paragraph, ref 107).

Response to comment 4: 

We have added a section “3.4 Limitations of the Liver Organoid Models” in section 3.

Reviewer 4 Report

This is a very interesting article on an emerging topic.

Some issues could improve the understanding of the article.

1. Figure 1-2, all abbreviations should be included in the figure caption. Each figure and table must be independent.

2. To visualize and specify each section of the review, the authors could incorporate tables with the main data. indicating the v advantages and disadvantages and its main uses (different cells, biomaterials, manufacturing methods, ....)

2. Figure 2 should be cited in the text.

3. review sentence of line 290

4. check for excessive use of "/"

Author Response

Response to comment 1:

The abbreviations of the components in the figure were included in the revised caption of Figure 1 and the figure was independently inserted in the paper.

Response to comment 2:

To visualize the development of bioengineering liver organoids, a time bar was illustrated in the revised Figure 1. The limitation of using liver organoids have been discussed in section 3.4 of the revised manuscript.

Response to comment 3:

Figure 2 was cited in the revised manuscript (see page 12 in the first paragraph).

Response to comment 4:

We have checked this sentence.

Response to comment 5:

We have checked all use of the "/".

Round 2

Reviewer 2 Report

The manuscript reads better in light of the revisions which have been undertaken

Reviewer 3 Report

The authors have modified the article and it is now good for publication. 

Reviewer 4 Report

The article is acceptable in its present form.